# Alcohol Consumption Norms and the Favored Alcohol Consumption Policies of Citizens of Seoul

**DOI:** 10.3390/ijerph21070834

**Published:** 2024-06-26

**Authors:** Jina Kang, Hyeri Han, Hyeongsu Kim, AhHyun Park, Dasom Kim, Rahil Hwang, Miyoung Kim, Young Ko, Sungwon Jung

**Affiliations:** 1Expert Group on Health Promotion for Seoul Metropolitan Government, Konkuk University, Seoul 05085, Republic of Korea; jinakang21@gmail.com (J.K.); hyeri2022@gmail.com (H.H.); ahpark.seoulhp@gmail.com (A.P.); dudurdaram@naver.com (D.K.); 2Department of Preventive Medicine, School of Medicine, Konkuk University, Seoul 05085, Republic of Korea; 3Department of Nursing, College of Nursing, Shinhan University, Dongducheon 11340, Republic of Korea; hwangri@hanmail.net; 4Department of Nursing, College of Nursing, Hanyang University, Seoul 04763, Republic of Korea; miyoung@hanyang.ac.kr; 5Department of Nursing, College of Nursing, Gachon University, Incheon 21936, Republic of Korea; youngko@gachon.ac.kr; 6Department of Nursing, Fareast University, Eumseong 27601, Republic of Korea; soora918@daum.net

**Keywords:** alcohol norms, alcohol consumption, alcohol policy

## Abstract

The purpose of this study was to define the alcohol consumption norms and attitudes toward alcohol regulation policies among citizens of Seoul and the relationships between such norms and the favored regulatory policies. The study population consisted of 1001 adults aged 19–80 years living in Seoul. We collected demographic data and data on health behaviors, attitudes towards drinking, and preferred alcohol regulation policies. The correlations between drinking and the favored regulatory policies were analyzed. Male, as well as being employed, aged 19–39 years, single, a smoker, and a current or heavy episodic drinker were associated with more positive attitudes toward drinking (all *p* < 0.001) and less desire for alcohol regulation policies (all *p* < 0.001). We found a significant negative correlation between attitudes toward drinking and preferred alcohol regulation policies (*p* < 0.001). Participants who favored reduced or no alcohol consumption and a reduction in alcohol-related harm were more accepting of restrictive alcohol consumption policies. To establish alcohol control polices, differences in drinking norms within populations should be considered. Furthermore, for a successful alcohol control policy, efforts should be made to change drinking norms, as well as consider differences in regulatory policy preferences between population groups.

## 1. Introduction

In South Korea, the annual alcohol consumption per adult has been decreasing over the past 10 years and reached the average level of all Organization for Economic Cooperation and Development (OECD) countries in 2020 [1]. However, the rate of adult high-risk drinking has increased over the past decade and was 12.4% in 2021 [2]. Moreover, the socioeconomic costs associated with excessive drinking exceeded 15 trillion KRW in 2019 [3]. The World Health Organization (WHO) has suggested 10 strategies that countries could implement to prevent excessive alcohol consumption and reduce alcohol related harm. Among these strategies, the five priority policies and interventions (SAFER) are as follows [4]: strengthen restrictions on alcohol availability; increase and enforce drunk driving countermeasures; facilitate the access of problem drinkers to screening, brief interventions, and treatment; enforce bans or comprehensive restrictions on alcohol advertising, sponsorship, and promotion; and raise alcohol prices by imposing excises and increasing prices.

If government interventions seeking to prevent excessive alcohol consumption and reduce alcohol-related harm are to be successful, the public must embrace norms that discourage drinking and minimize drinking harm regardless of individual drinking status. A conventional social norm (including a drinking norm) is a normative social belief, i.e., how an individual views the behaviors and attitudes of others in a social setting. Therefore, this is a cognitive construct (a mental representation) of an actual social norm [5]. Changes in a social norm equate to variations in behaviors or beliefs; changes in either of these domains are not necessarily accompanied by changes in the other [5]. 

Most studies on the alcohol regulation policies of South Korea have evaluated the implementation status of the 10 strategies suggested by the WHO [6] and/or considered how international trends and policies that sought to reduce the harm caused by alcohol consumption could inform national policy [7]. One study compared various national policies aimed at regulating alcohol consumption in public places [8]. However, few authors have explored the drinking norms of alcohol consumers or the general public, or their attitudes toward alcohol regulation policies [9]. This study describes the drinking norms and preferences for alcohol regulation policies among citizens of Seoul and explores the relationships between these two variables. The goal was to show how drinking norms are formed and to identify alcohol regulation policies that might find general favor.

## 2. Materials and Methods

### 2.1. Study Design, Data, and Population

We analyzed cross-sectional survey data on the drinking norms and policy preferences of citizens of Seoul; the survey was conducted by the “Smart Health Promotion Department” of the Seoul Metropolitan Government. The study population was selected using a proportional stratified sampling method based on residence (across the 25 districts of Seoul), gender, and age. Sampling was conducted using an online panel created by the “M Survey Company”; all participants were aged 19–80 years. In total, 1001 individuals were sampled. Data were collected from 9 to 15 February 2022.

### 2.2. Variables

#### 2.2.1. Sociodemographic Characteristics

The sociodemographic variables recorded included gender, age, marital status, highest educational level, average monthly household income, employment status, and the number of cohabiting family members. Age was categorized as 19–39, 40–59, or ≥60 years. Marital status was classified as unmarried, married, or other (divorced, widowed, or separated). Educational level was categorized as high school graduate or below or college graduate or above (attended graduate school). The monthly household income was divided into <2 million KRW, 2–2.99 million KRW, and ≥3 million KRW. Employment status was categorized as working or not. “Cohabiting family member” status was divided into living alone or living with more than two people.

#### 2.2.2. Health Behavior Variables

Health behavior variables included alcohol consumption (current drinkers, past 12 months abstainers, and heavy episodic drinkers) and current smoking status. Current drinkers were defined as people who have consumed alcoholic beverages in the previous 12-month period. Past 12 months abstainers were defined as people who did not drink any alcohol in the previous 12-month period. This includes former drinkers (people who have previously consumed alcohol but who have not done so in the previous 12-month period) and lifetime abstainers. Heavy episodic drinking is often defined in terms of exceeding a certain daily volume or quantity per occasion, or daily drinking [10]. In Korea, heavy episodic drinking was defined as alcohol consumption more than twice a week, with an average of seven drinks for males and five drinks for females at any time. Smoking was categorized as yes or no.

#### 2.2.3. Drinking Norms

Drinking norms were defined using a survey that explored attitudes toward alcohol in the study of Son et al. [9]. The survey included eight statements, and the respondents’ attitudes toward the statements were scored using a five-point Likert scale, ranging from strongly disagree to strongly agree. The possible range of total scores for the 8 items of drinking norms is 8–40. Higher scores indicated more positive attitudes toward the statements, which were as follows: ① It is acceptable to drink alcohol in parks or in mountain reserves; ② It is acceptable to drink alcohol during the daytime; ③ It is acceptable to get drunk; ④ It is acceptable that minors drink alcohol; ⑤ Drinking alone is fine; ⑥ Refusing alcohol offered by another drinker is impolite; ⑦ Drinking alcohol at benches outside convenience stores is acceptable; and, ⑧ Reduced punishments for crimes committed under the influence of alcohol are justifiable. The Cronbach’s alpha coefficient for the eight statements that measured attitudes toward drinking was 0.818.

#### 2.2.4. Alcohol Policy Preferences

The preferred alcohol policies were evaluated using a survey (11 questions) derived from Son et al. [9]. The survey covered alcohol price limits, alcohol use policies, and marketing and advertising limitations. Attitudes were explored by recording the reactions to various statements. Statements pertaining to price restriction policies were as follows: “the price of alcohol should be increased above the current level” and “promotional activities such as discount sales should be restricted”. Statements relating to usage restriction suggested “expanding designated non-drinking areas in public places”, “restricting alcohol consumption during gatherings or events”, “strengthening restrictions on locations where alcohol is sold”, and “limiting the hours of alcohol sales”. Statements relating to marketing restriction policies suggested that “alcohol should have warning labels like cigarettes”, and “free alcohol donated by alcohol companies during local events or festivals should be forbidden”. Statements relating to attitudes toward advertising restrictions suggested that “alcohol advertisements featuring celebrities or entertainers should be prohibited”, “drinking scenes should not be shown on visual media (TV, YouTube, or social media)” and “alcohol advertisements on visual media should be restricted”. Attitudes toward alcohol regulation policies were measured on a 4-point Likert scale ranging from strongly disagree to strongly agree. The possible range of total scores of alcohol policy preferences is 11–44 for the 11 items, with higher score indicating greater preference for regulation in each area.

### 2.3. Data Analysis

All data analyses were conducted using IBM SPSS for Windows software (ver. 27.0; IBM Corp., Armonk, NY, USA). The demographic characteristics and health behaviors of the study population are presented as frequencies with percentages. Attitudes toward drinking and preferences in terms of alcohol regulation policies are presented as means with standard errors. The views and preferences were analyzed according to demographic characteristics and health behaviors. Comparisons were performed using the chi-squared test and ANOVA, as appropriate. Pearson correlation coefficients between attitudes toward drinking and the favored policies were derived. The level of statistical significance was set to *p* < 0.05.

## 3. Results

### 3.1. General Characteristics of the Study Population

Of all respondents, 51.4% (*n* = 515) were female, 55.1% (*n* = 552) were married, 75.0% (*n* = 751) were college graduates or above, 73.6% (*n* = 737) were workers, and 64.5% (*n* = 646) had a monthly household income ≥3 million KRW (Table 1). The current drinkers rate was 77.8%, and the past 12 months abstainers rate was 22.2%. 

### 3.2. Attitudes toward Drinking

The mean (standard error) score for the eight questions measuring attitudes toward drinking was 2.27 (0.02) (Table 2). The item “Drinking alone is fine” had the highest score of 3.22 (0.03), followed by “It is acceptable to drink alcohol during the day”, at 2.61 (0.03), and “It is acceptable to get drunk” at 2.59 (0.04). In contrast, “Reducing punishment for crimes committed under the influence of alcohol is justifiable” had the lowest score, of 1.40 (0.03), followed by “It is acceptable for minors to drink alcohol” at 1.61 (0.03). 

Table 3 shows the total scores for the eight scenarios according to the general population characteristics. The overall score was significantly higher for male (2.38) than female (2.17) (*p* < 0.001). Participants aged 19–39 years scored significantly higher (2.45) than those aged 40–59 years (2.22) and those aged > 60 (2.09) years (both *p* < 0.001). Single individuals had a score of 2.39, significantly higher than married persons (2.18) (*p* < 0.001). Those who were employed had a score of 2.33, significantly higher than that of those who were unemployed (2.11) (*p* < 0.001). In terms of health behaviors, smokers scored significantly higher (2.57) than non-smokers (2.19) (*p* < 0.001). Similarly, current drinkers (score = 2.33) and heavy episodic drinkers (2.54) scored significantly higher than the other participants (both *p* < 0.001).

The results for the eight questions that evaluated drinking perspectives are presented in Appendix A according to the general population characteristics.

### 3.3. Preferred Alcohol Regulation Policies

The mean (standard error) preference score for the four types (11 items) of alcohol regulation policies was 2.81 (0.02) (Table 4). Among the four policies, “Advertising media restriction” had the highest score of 2.98 (0.02) followed by “Marketing restriction” at 2.97 (0.02), “Usage restriction” at 2.89 (0.02), and “Price restriction” at 2.39 (0.03). Among the 11 items, “Alcohol restriction during gatherings/events” had the highest score of 3.17 (0.03), followed by “Expansion of non-drinking zones in public places” at 3.16 (0.03) and “Restrictions on alcohol advertising” at 3.03 (0.03). Conversely, “Increased alcohol prices” had the lowest score of 2.34 (0.03) followed by “Restrictions on alcohol promotion activities” at 2.44 (0.03).

The preference scores for the four alcohol regulation policies according to the general characteristics of the study population are shown in Table 5. Female, participants aged ≥60 years, those with a high school education or less, multifamily households, non-smokers, and past 12 months abstainers or low-risk drinkers exhibited significantly higher preferences than the other participants for the “Price restriction policy”. Female, participants ≥ 40 years of age, married individuals, unemployed persons, non-smokers, and past-12 months abstainers or low-risk drinkers had a significantly greater preference for the “Usage restriction policy” than the other participants. Female, participants ≥ 40 years of age, married individuals, non-smokers, and past 12 months abstainers or low-risk drinkers showed a significantly higher preference for the “Marketing restriction policy” than the other participants. Finally, female, participants ≥ 40 years of age, married individuals, non-smokers, and past 12 months abstainers exhibited a significantly higher preference for the “Advertising restriction policy” than the other participants.

The preferences for the 11 alcohol regulation policies are shown in Appendix A according to the general characteristics of the study population.

### 3.4. Correlations between Attitudes toward Drinking and Preferred Alcohol Regulation Policies

The scores reflecting attitudes toward drinking and preferences for the four types of drinking regulation policies exhibited significant negative correlations (all *p* < 0.001, Table 6). Specifically, the correlation coefficient between the scores for attitudes toward drinking and preference for alcohol usage restriction policies was –0.426 (*p* < 0.001). Also, we found significant positive correlations among the preferences for the four types of alcohol regulation policies (*p* < 0.001), particularly between usage and marketing restriction policies (r = 0.725, *p* < 0.001).

The correlations between the scores for the eight questions that measured attitudes toward drinking and those for the 11 items assessing the preferred alcohol regulation policies are shown in Appendix A.

## 4. Discussion

This study explored existing drinking norms, including how they formed and how they affected the development of alcohol regulation policies. We analyzed correlations between the drinking norms of citizens of Seoul and their preferred alcohol regulation policies.

The mean score of Seoul citizens for attitudes toward drinking (on a scale of 1–5) was 2.27, which was lower than the median score of 2.5. Of the eight scenarios, “drinking alone” and “drinking during daytime” were relatively well-accepted. In contrast, attitudes toward “minors drinking” and “reducing penalties for crimes committed under the influence of alcohol” were relatively negative. Generally, drinking is associated with gathering in a pub with others in the evening after the day’s work concludes [11]. However, changes in industrial structures and shift work, the increased number of single-person households, and/or a reluctance to interact with others are reinforcing a “solo culture”. The fact that attitudes toward “drinking alone” and “daytime drinking” were relatively positive suggests that these practices are increasingly becoming accepted as drinking norms in Korea [12]. On the other hand, the very negative attitude toward “minors drinking” indicates that our society considers that the need for the healthy physical and mental growth of minors makes exposure to alcohol inappropriate. Excess alcohol consumption plays a major role in traffic accidents and assaults [13]. Some have suggested that the severity of punishments should be reduced because the self-control of perpetrators is lessened by alcohol. Others have called for more stringent punitive measures to prevent similar future incidents [14]. The highly negative attitude of Seoul citizens toward “reducing penalties for crimes committed under the influence of alcohol” reflects the latter perspective. Thus, the societal drinking norm is that punishments for crimes committed under the influence should not be reduced; this may reduce the number of future accidents caused by drinking.

When the attitudes toward drinking were examined according to sociodemographic characteristics and health behaviors, participants who were more exposed to drinking and exhibited higher rates of alcohol consumption had more positive attitudes toward alcohol. The former participants included males, employed persons, those aged 19–39 years, singletons, smokers, and heavy episodic drinkers. The results of this study align with the previous finding that “frequent heavy drinkers believed that heavy alcohol use is more normative in social reference groups compared with lighter drinkers” [15].

Among national-level strategies designed to restrict alcohol use and reduce alcohol-related harm, price control policies such as higher alcohol prices and limited discounts [16], usage restriction policies such as reduced sales times and fewer places where alcohol can be consumed, and the designation of alcohol-free zones have significantly reduced alcohol consumption [17]. Conversely, marketing restriction policies such as the attachment of warning notices and information/education campaigns have been less effective [17,18]. Among the five strategies proposed by the WHO [4], those that have been widely implemented in Korea include measures that reduce drunk driving and alcohol usage (an age limit for alcohol purchases is in place). Policies that are being implemented but require improvement include alcohol usage restrictions (the prohibition of public drinking) and regulations on alcohol marketing, pricing policies, and policies that reduce alcohol-related harm. To date, alcohol sales have not been restricted [19]. In this study, the most-preferred alcohol regulation policies were advertising restrictions and marketing limitations; usage restrictions and price controls were less favored. This aligns with other research findings [19] that suggest greater resistance among both drinkers and the general population toward policies that significantly reduce alcohol consumption; initiatives that affect alcohol consumption to a lesser extent are preferred.

Our participants who were less exposed to alcohol, including females, individuals ≥ 40 years of age, married persons (regardless of whether the spouse was present or not), those educated up to high school level, unemployed individuals, past 12 months abstainers, and non-heavy episodic drinkers, showed a higher preference for policies that would reduce alcohol consumption. Many studies from Australia, North America, and Europe have shown that females, the elderly, and past 12 months abstainers tend to express a higher preference for alcohol regulation policies [20,21,22]. Similarly, Korean studies have found that females, older individuals, married people, those with lower educational levels, and non-heavy episodic drinkers have more favorable attitudes toward alcohol regulation policies [23]. In studies that explored the attitudes toward alcohol regulation policies of problem drinkers and normal drinkers, the former preferred policies aiming to reduce drinking-related harm, such as the stricter enforcement of drunk driving laws [24]. On the other hand, normal drinkers tended to prefer policies that prevented drinking, i.e., usage restriction or price control initiatives [23].

Finally, we found significant negative correlations between scores for attitudes toward drinking and the preferred alcohol regulation policies. In other words, the more positive the attitude toward drinking, the lower the preference for alcohol regulation policies, especially price controls. If alcohol regulation policies aiming to prevent drinking or reduce alcohol-related harm are to be successfully implemented, it is imperative that population drinking norms align with drinking prevention and reduced drinking harm.

Many prior studies have shown that drinking norms have been linked to drinking culture and have a significant impact on alcohol consumption [25,26,27]. Based on these findings, studies have shown that an intervention program developed to change one’s understanding of drinking norms can successfully reduce alcohol consumption [28,29].

The limitations of this study included the fact that attitudes toward drinking and alcohol restriction policies were self-reported. Such data can be gathered in various ways. However, the tools used to evaluate attitudes in this study are also employed in research conducted by the Ministry of Health and Welfare, and they have been confirmed to yield objectively valid data [9]. Also, the alcohol regulation policies that we explored did not include strategies that might reduce alcohol-related harm, such as stricter regulations on drunk driving or interventions or treatments for drinkers. Future research utilizing various tools to explore drinking norms and attitudes toward alcohol restriction policies will overcome these limitations. Finally, all participants were citizens of Seoul; the generalization of the results to other Koreans should be performed with caution. However, we suggest that regional differences will not prohibit the use of our results to create appropriate drinking norms.

In conclusion, population groups need to have drinking norms oriented towards alcohol prevention and harm reduction to increase the acceptance of restrictive alcohol policies. Therefore, continuous societal efforts are necessary to help groups with a positive attitude toward drinking in order for them to adopt desirable drinking norms.

## Figures and Tables

**Table 1 ijerph-21-00834-t001:** Participant sociodemographic characteristics and health behaviors.

Characteristics		Current Drinker, *n* (%)	Past 12 Months Abstainer *n* (%)	Total, *n* (%)
Total		779 (77.8)	222 (22.2)	1001 (100.0)
Gender	Female	370 (71.8)	145 (28.2)	515 (51.4)
	Male	409 (84.2)	77 (15.8)	486 (48.6)
Age (years)	19~39	303 (82.8)	63 (17.2)	366 (36.6)
	40~59	301(79.0)	80 (21.0)	381 (38.1)
	≥60	175 (68.9)	79 (31.1)	254 (25.4)
Marital status	Unmarried	315 (81.6)	71 (18.4)	386 (38.6)
	Married	431 (78.1)	121 (21.9)	552 (55.1)
	Other (divorced, separated, widowed)	33 (52.4)	30 (47.6)	63 (6.3)
Educational status	≤High school	163 (65.2)	87 (34.8)	250 (25.0)
	≥College, university	616 (82.0)	135 (18.0)	751 (75.0)
Job status	Employed	588 (79.8)	149 (20.2)	737 (73.6)
	Unemployed	191 (72.3)	73 (27.7)	264 (26.4)
Monthly household income (KRW)	<2,000,000	91 (63.2)	53 (36.8)	144 (14.4)
	2,000,000–2,999,999	164 (77.7)	47 (22.3)	211 (21.1)
	≥3,000,000	524 (81.1)	122 (18.9)	646 (64.5)
Number of household members	Living alone	95 (72.5)	36 (27.5)	131 (13.1)
	≥2	684 (78.6)	186 (21.4)	870 (86.9)
Smoking status	Non-smoker	588 (74.1)	205 (25.9)	793 (79.2)
	Current smoker	191 (91.8)	17 (8.2)	208 (20.8)

**Table 2 ijerph-21-00834-t002:** Drinking norm scores for each item of all participants.

Drinking Norms ^1^	Mean (SE)
It is okay to drink alone	3.22 (0.03)
It is okay to drink alcohol in the daytime	2.61 (0.03)
It is okay to get a little tipsy	2.59 (0.04)
It is okay to drink at a convenience store (under internal or external parasols)	2.54 (0.03)
It is discourteous to refuse a drink offered by someone else	2.14 (0.03)
It is okay to drink in a park or a mountain reserve	2.07 (0.03)
It is okay for high school students to drink	1.61 (0.03)
It is okay to lower the punishments for crimes committed after drinking alcohol	1.40 (0.03)
Total	2.27 (0.02)

^1^ Data are five-point Likert scale scores.

**Table 3 ijerph-21-00834-t003:** Total outcome of drinking norms according to sociodemographic characteristics and health behaviors.

Characteristics		Total Outcome of Drinking Norms Scale Mean (SE)	t/F	*p*	Scheffe
Total		2.27 (0.02)			
Gender	Female	2.17 (0.02)	−4.837	0.000	
	Male	2.38 (0.03)			
Age (years)	19~39 ^a^	2.45 (0.05)	25.671	0.000	
	40~59 ^b^	2.22 (0.05)			c < b < a *
	≥60 ^c^	2.09 (0.05)			
Marital status	Unmarried ^a^	2.39 (0.03)	11.062	0.000	
	Married ^b^	2.18 (0.03)			b < a *
	Other (divorced, separated, widowed) ^c^	2.33 (0.09)			
Educational status	≤High school	2.22 (0.05)	−1.267	0.206	
	≥College, university	2.29 (0.02)			
Job status	Employed	2.33 (0.03)	4.519	0.000	
	Unemployed	2.11 (0.04)			
Monthly household income (KRW)	<2,000,000	2.20 (0.06)	1.763	0.172	
	2,000,000–2,999,999	2.34 (0.05)			
	≥3,000,000	2.27 (0.03)			
Number of household members	Living alone	2.28 (0.06)	0.223	0.824	
	≥2	2.27 (0.02)			
Smoking status	Non-smoker	2.19 (0.02)	−6.759	0.000	
	Current smoker	2.57 (0.05)			
Current drinker	No	2.06 (0.05)	−5.355	0.000	
	Yes	2.33 (0.02)			
Heavy episodic drinker	No	2.24 (0.02)	−3.641	0.000	
	Yes	2.54 (0.09)			

* *p* < 0.001. Superscrips a,b and c indicate a group for a post hoc test.

**Table 4 ijerph-21-00834-t004:** Total scores for each item of attitudes toward alcohol control policies.

Dimension	Item ^1^	Mean (SE)
Pricing		2.39 (0.03)
	Liquor price increases	2.34 (0.03)
	Restrictions on liquor promotion activities	2.44 (0.03)
Restrictions on utilization		2.89 (0.02)
	Restrictions on places that sell alcoholic beverages	2.88 (0.03)
	Restrictions on liquor selling times	2.63 (0.03)
	Expansion of drinking prohibitions in public places	3.16 (0.03)
	Restrictions on drinking at rallies (events)	3.17 (0.03)
Marketing restrictions		2.97 (0.02)
	Attach warning images to bottles	2.99 (0.03)
	Restrictions on the provision of free alcoholic beverages at local events	2.74 (0.03)
Restrictions on advertising		2.98 (0.02)
	Restrictions on liquor advertisements	3.03 (0.03)
	Restrictions on drinking scenes in visual media	2.94 (0.03)
	Restrictions on liquor advertisements in visual media	2.97 (0.03)
Total		2.81 (0.02)

^1^ Data are four-point Likert scale scores.

**Table 5 ijerph-21-00834-t005:** Preferred alcohol control policies according to sociodemographic characteristics and health behaviors.

Characteristics		Dimensions of the Alcohol Control Policies							
		Pricing		Restrictions on Utilization		Marketing Restrictions		Restrictions on Advertising Media	
		Mean (SE)	t/F(p)	Mean (SE)	t/F(p)	Mean (SE)	t/F(p)	Mean (SE)	t/F(p)
Total		2.39 (0.03)		2.89 (0.02)		2.97 (0.02)		2.98 (0.02)	
Gender	Female	2.48 (0.04)	3.425 (0.001)	2.99 (0.03)	4.539 (0.000)	3.09 (0.03)	5.766 (0.000)	3.10 (0.03)	5.069 (0.000)
	Male	2.30 (0.04)		2.79 (0.03)		2.84 (0.03)		2.85 (0.04)	
Age (years)	19~39 ^a^	2.30 (0.04)	4.533 (0.011) a < c ^1^	2.74 (0.04)	14.928 (0.000) a < b,c ^2^	2.86 (0.04)	7.452 (0.001) a < b,c ^1^	2.84 (0.04)	8.055 (0.000) a < b,c ^2^
	40~59 ^b^	2.43 (0.04)		2.95 (0.03)		3.04 (0.03)		3.04 (0.04)	
	≥60 ^c^	2.48 (0.05)		3.03 (0.04)		3.01 (0.04)		3.07 (0.05)	
Marital status	Unmarried ^a^	2.32 (0.04)	3.664 (0.026)	2.78 (0.04)	9.063 (0.000) a < b ^2^	2.89 (0.04)	4.003 (0.020) a < b ^2^	2.86 (0.04)	6.642 (0.002) a < b ^2^
	Married ^b^	2.43 (0.03)		2.98 (0.03)		3.02 (0.03)		3.06 (0.03)	
	Other (divorced, separated, widowed) ^c^	2.57 (0.10)		2.850 (0.08)		2.94 (0.08)		2.98 (0.09)	
Educational status	≤High school	2.56 (0.05)	3.818 (0.000)	2.95 (0.04)	1.662 (0.097)	2.99 (0.05)	0.571 (0.568)	3.03 (0.05)	1.270 (0.204)
	≥College, university	2.34 (0.03)		2.87 (0.03)		2.96 (0.02)		2.96 (0.03)	
Job status	With job	2.39 (0.03)	−0.174 (0.862)	2.86 (0.03)	−2.372 (0.018)	2.95 (0.03)	−1.581 (0.114)	2.97 (0.03)	−0.725 (0.469)
	No job	2.40 (0.05)		2.98 (0.04)		3.02 (0.04)		3.01 (0.05)	
Household Income (KRW)	<2,000,000	2.42 (0.07)	0.390 (0.677)	2.90 (0.06)	1.769 (0.171)	2.96 (0.06)	0.391 (0.676)	3.02 (0.07)	0.259 (0.772)
	2,000,000–2,999,999	2.35 (0.05)		2.81 (0.05)		2.93 (0.05)		2.96 (0.06)	
	≥3,000,000	2.40 (0.03)		2.91 (0.03)		2.98 (0.03)		2.98 (0.03)	
Number of household members	Living alone	2.24 (0.06)	−2.680 (0.008)	2.84 (0.06)	−0.937 (0.349)	2.96 (0.06)	−0.126 (0.900)	2.92 (0.07)	−0.969 (0.333)
	≥2	2.42 (0.03)		2.90 (0.02)		2.97 (0.02)		2.99 (0.03)	
Smoking status	Non-smoker	2.43 (0.03)	2.656 (0.008)	2.95 (0.02)	5.047 (0.000)	3.03 (0.02)	5.886 (0.000)	3.04 (0.03)	4.880 (0.000)
	Current smoker	2.26 (0.06)		2.67 (0.05)		2.72 (0.05)		2.75 (0.06)	
Current drinker	No	2.91 (0.05)	11.346 (0.000)	3.18 (0.04)	7.210 (0.000)	3.21 (0.04)	6.106 (0.000)	3.19 (0.05)	4.722 (0.000)
	Yes	2.25 (0.03)		2.81 (0.02)		2.90 (0.02)		2.92 (0.03)	
Heavy episodic drinker	No	2.37 (0.03)	4.251 (0.000)	2.89 (0.02)	2.617 (0.009)	2.98 (0.02)	2.642 (0.010)	2.98 (0.03)	1.926 (0.054)
	Yes	1.94 (0.10)		2.66 (0.11)		2.69 (0.11)		2.78 (0.11)	

^1^ Scheffe test; ^2^ Dunnett test. Superscrips a, b and c indicate a group for a post hoc test.

**Table 6 ijerph-21-00834-t006:** Correlations between drinking norms and preferences for alcohol control policy dimensions.

Variables	1 r(*p)*	2 r(*p)*	3 r(*p)*	4 r(*p)*	5 r(*p)*
1. Mean of drinking norms	1				
2. Policy preferences regarding price limits	−0.273 (0.000)	1			
3. Policy preferences regarding usage restrictions	−0.426 (0.000)	0.591 (0.000)	1		
4. Policy preferences regarding marketing restrictions	−0.407 (0.000)	0.521 (0.000)	0.725 (0.000)	1	
5. Policy preferences regarding advertising media restrictions	−0.351 (0.000)	0.430 (0.000)	0.611 (0.000)	0.683 (0.000)	1

## Data Availability

The datasets used and/or analyzed during the current study are available available from the corresponding author upon reasonable request and with permission from the “Smart Health Promotion Department” of the Seoul Metropolitan Government.

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
