# Peer review of "Alcohol Consumption Norms and the Favored Alcohol Consumption Policies of Citizens of Seoul"

_ijerph, 2024, doi:10.3390/ijerph21070834_

Round 1

Reviewer 1 Report

Comments and Suggestions for Authors

Dear Authors,

Your manuscript include the interesting and importand the information conected with promotion of the health life style, The manuscript results may conduct to creation of alcohol consumption barriers and it is very inportant in the limitation to access of alcohol, but the text should to be corrected according to the suggestions:

Line 75:

It is: 20–39, 40–59, or > 60

It should be: 19–39, 40–59, or  60

Line 79:

It is: < 2 million Won, 2–2.99 million Won, and > 3 million Won

It should be: < 2 million Won, 2–2.99 million Won, and  3 million Won.

Lines 82-86

"annual or high-risk drinking" should correspond to WHO recommendation and this part of manuscript should include of references, for example according to AUDIT etc.

Line 88 - "Drinking norms"

It is necessary in the detail describing of "drinking norms scale", for example a range of outcome and a total outcome, because this data the authors use in the tables 2 and 3. And similar it is necessary describing of "alcohol control policies".

Line 132

Is it: income > 3 million Won

It should be: income  3 million Won

Line 136 (table 1)

The sentence "Non-annual drinker" should be explained and clarification in the part of material and methods. Reader does not know what is it "Non-annual drinker".

Line 158

In the column of "Mean (SE)" it is necessary to add sentence "Total outcome of Drinking norms scale". 

Lines 128-202

Each tables have to has the references of table in the text for example "(Table 3)" before or below the table. And each tables have to has a description their contents. Authors should check this and to correct. Reader may be bit confused in this part of manuscript.

Line 227-264

This text should be at the beginning of Discussion part , according to order of result part .

Line 282

I suggest extraction of conclusion part at the end of text.

Best regars

Reviewer

Author Response

Response to Reviewer 1’ comments

Thanks for your good critics to improve the quality of our manuscript.

Using your critics, we corrected as below.

  1. Comment on the correction of words (lines 75, 79, 132)

Line 75: It is: 20–39, 40–59, or > 60,  It should be: 19–39, 40–59, or ≥ 60

Line 79: It is: < 2 million Won, 2–2.99 million Won, and > 3 million Won

It should be: < 2 million Won, 2–2.99 million Won, and ≥ 3 million Won.

Line 132 Is it: income > 3 million Won  It should be: income ≥ 3 million Won

Response to the comment

According to the reviewer’s suggestion, we corrected some symbols and words as follows.

Line 75 : 19–39, 40–59, or ≥ 60

Line 79 : ≥ 3 million Won.

Line 132: income ≥ 3 million Won

  1. Comment on the study terms, ‘annual drinker, non-annual drinker, high-risk drinker’

Line 82-86: "annual or high-risk drinking" should correspond to WHO recommendation and this part of

manuscript should include of references, for example according to AUDIT etc.

Line 136 (table 1): The sentence "Non-annual drinker" should be explained and clarification in the part

of material and methods. Reader does not know what is it "Non-annual drinker"

Response to the comment

According to the reviewer’s suggestion, we modified the study terms and clarified the definition based on the WHO references. ‘Annual drinker’ was modified by ‘current drinker’, ‘non-annual drinker’ was modified by ‘past-12month abstainer’ and ‘high-risk drinker’ was modified as ‘heavy episodic drinker’ as follows. Furthermore, these terms were applied throughout the manuscript and tables.

#Original contents (lines 82-86)

Annual drinking was considered present when a respondent drank alcohol even once in the past year and high-risk drinking was defined as alcohol consumption more than twice a week, with an average of seven drinks for males and five drinks for females at any time.

#Revised contents(lines 85-94, table1,3,5)

Health behavior variables included alcohol consumption (current drinker, past-12 month abstainer, and heavy episodic drinker) and current smoking status. Current drinkers were defined as people who have consumed alcoholic beverages in the previous 12-month period. Past-12 month abstainers were defined as people who did not drink any alcohol in the previous 12-month period. This includes former drinkers(people who have previously consumed alcohol but who have not done so in the previous 12-month period) and lifetime abstainers. And heavy episodic drinking is often defined in terms of exceeding a certain daily volume or quantity per occasion, or daily drinking [9]. In Korea, heavy episodic drinking was defined as alcohol consumption more than twice a week, with an average of seven drinks for males and five drinks for females at any time. *Reference: World Health Organization. Global status report on alcohol and health 2018. World Health Organization, 2019. ISBN 978-92-4-156563-9.

  1. Comment on the ‘table3’

Line 158: In the column of "Mean (SE)" it is necessary to add sentence "Total outcome of Drinking norms scale".

Response to the comment

According to the reviewer’s suggestion about line 158, we add the sentence “Total outcome of Drinking norms scale” in the table 3 (revised line170)

  1. Comment for ‘checking the description of each table’

Lines 128-202: Each tables have to has the references of table in the text for example "(Table 3)" before or below the table. And each tables have to has a description their contents. Authors should check this and to correct. Reader may be bit confused in this part of manuscript.

Response to the comment

According to the reviewer’s suggestion, we checked and corrected the position of one table. Table 2 was moved into line154 which was below the explanation of table2. Table 4 was moved into line 182 which was below the explanation of table4. And we checked and confirmed that there were no problems in other tables.

  1. Comment on the ‘Drinking norms scale’ and the ‘alcohol control policies scale’

Line 88-“Drinking norms” : It is necessary in the detail describing of "drinking norms scale", for example a range of outcome and a total outcome, because this data the authors use in the tables 2 and 3. And similar it is necessary describing of "alcohol control policies".

Response to the comment

According to the reviewer’s suggestion, we added the explanation on ‘drinking norms scale’ and ‘alcohol control policies’ as follows.

The survey included eight statements, and the respondents’ attitudes toward the statements were scored using a five-point Likert scale, ranging from strongly disagree to strongly agree. The possible range of total scores for the 8 items of drinking norms is 8-40.  (revised lines 99-100).         

Attitudes toward alcohol regulation policies were measured on a 4-point Likert scale ranging from strongly disagree to strongly agree. The possible range of total scores of alcohol policy preferences is 11-44 for the 11 items, with higher score indicating greater preference for regulation in each area (revised lines 126-128).

  1. Comment regarding the order of discussion part

Line 227-264 : This text should be at the beginning of Discussion part, according to order of result part .

Response to the comment

Actually, we describe according to order of the result part. The first paragraph(revised line 240-246) presents discussion about table 3(attitudes toward drinking according to sociodemographic characteristics and health behaviors) and the second paragraph(revised lines 247-263) discusses  attitudes toward alcohol control policies (table 4,5,6).

  1. Comment on ‘conclusion part’

Line 282 : I suggest extraction of conclusion part at the end of text.

Response to the comment

According to the reviewer’s suggestion, we extracted the conclusion at the end of text as follows.

“In conclusion, population groups need to have drinking norms oriented towards alcohol prevention and harm reduction to increase acceptance of restrictive alcohol policies. Therefore, continuous social efforts are necessary to help groups with a positive attitude toward drinking in order for them to adopt desirable drinking norms.” (revised line 300-303)

Reviewer 2 Report

Comments and Suggestions for Authors

In this review of the empirical manuscript titled "Alcohol Consumption Norms and the Favored Alcohol Consumption Policies of Citizens of Seoul" I found interesting findings and recommendations for next steps. The authors should be commended for their innovative and exciting work that will be of great interest to a broad readership. few studies if any have researched this topic which is of great interest to policy makers and those working in alcohol prevention. 

I found a few areas where the manuscript could be enhanced or where clarifications may be needed.

1. The conclusion in the abstract is not thoughtful and should be revised. And, there is no need to list the number of survey items in the abstract. 

2. There are a few spelling errors and awkward prepositions etc so suggest a minor readability check to ensure those are fixed.

3. It was not clear who the non annual drinkers are? This should be presented more clearly.

4. Table 5 is a bit confusing and challenging to read, because it is compressed in to portrait format.

5. Take a advantage of guiding the reader by providing more descriptive details in Table titles. 

6. The paper could be enhanced by international comparisons, particularly in the discussion. There is significant movement in policy making. And, great success stories available. What lessons learned could be beneficial to Seoul?

7. There are relatively few citations in the intro and discussion. In discussion, even sentence on alcohol harm has no citations. This needs to be addressed throughout.

8. The manuscripts concludes with the limitations. Please consider adding something more thoughtful about next steps, how the data will be used or any expectations.

Overall, I found the manuscript easy to follow, analyses clearly presented and of great interest. Most importantly, the topic is of great interest. 

Comments on the Quality of English Language

A few sentences here and there can be improved. Pay particular attention to prepositions. 

Author Response

Response to Reviewer 2’ comments

Thanks for your good critics to improve the quality of our manuscript.

Using your critics, we corrected as below.

  1. Comment on the ‘abstract’

The conclusion in the abstract is not thoughtful and should be revised. And, there is no need to list the number of survey items in the abstract.

Response to the comment

According to the reviewer’s comment, we revised the conclusion of abstract and deleted the number of survey items as follows.

“Abstract: The purpose of this study was to define the alcohol consumption norms and attitudes toward alcohol regulation policies among citizens of Seoul, and the relationships be-tween such norms and the favored regulatory policies. The study population consisted of 1,001 adults aged 19–80 years living in Seoul. We collected demographic data and data on health behaviors, attitudes towards drinking, and preferred alcohol regulation policies. The correlations between drinking and the favored regulatory policies were analyzed. Male, as well as being employed, aged 19–39 years, single, a smoker, and a current or heavy episodic drinker were associated with more positive attitudes toward drinking (all p < 0.001) and less desire for alcohol regulation policies (all p < 0.001). We found a significant negative correlation between attitudes toward drinking and preferred alcohol regulation policies (p < 0.001). Participants who favored reduced or no alcohol consumption and a reduction of alcohol related harm were more accepting of restrictive alcohol consumption policies. To establish alcohol control polices, differences in drinking norms within populations should be considered. Furthermore, for a successful alcohol control policy, efforts should be made to change drinking norms, as well as consider differences in regulatory policy preferences between population groups”

  1. Comment on the ‘readability checking’

There are a few spelling errors and awkward prepositions etc. so suggest a minor readability check to ensure those are fixed.

Response to the comment

 According to the reviewer’s suggestion, we carefully checked and corrected the spelling and grammar with the help of a colleague proficient in English writing. 

  1. Comment on the study terms ‘annual drinker’

It was not clear who the non annual drinkers are? This should be presented more clearly.

 Response to the comment

According to the reviewers’ suggestion, we modified the several main study terms and clarified the definition based on the WHO references. ‘Annual drinker’ was modified by ‘current drinker’, ‘non-annual drinker’ was modified by ‘past-12month abstainer’ and ‘high-risk drinker’ was modified as ‘heavy episodic drinker’ as follows. Furthermore, these terms were applied throughout the manuscript and tables.

#Original contents(lines 82-86)

Annual drinking was considered present when a respondent drank alcohol even once in the past year and high-risk drinking was defined as alcohol consumption more than twice a week, with an average of seven drinks for males and five drinks for females at any time.

#Revised contents(lines 85-94, tabel1,3,5)

Health behavior variables included alcohol consumption (current drinker, past-12 month abstainer, and heavy episodic drinker) and current smoking status. Current drinkers were defined as people who have consumed alcoholic beverages in the previous 12-month period. Past-12 month abstainers were defined as people who did not drink any alcohol in the previous 12-month period. This includes former drinkers(people who have previously consumed alcohol but who have not done so in the previous 12-month period) and lifetime abstainers. And heavy episodic drinking is often defined in terms of exceeding a certain daily volume or quantity per occasion, or daily drinking [9]. In Korea, heavy episodic drinking was defined as alcohol consumption more than twice a week, with an average of seven drinks for males and five drinks for females at any time. *Reference: World Health Organization. Global status report on alcohol and health 2018. World Health Organization, 2019. ISBN 978-92-4-156563-9.

  1. Comment on the ‘table5’

Table 5 is a bit confusing and challenging to read, because it is compressed in to portrait format.

Response to the comment

Reflecting the reviewer’s suggestion, we changed the format of table 5 to horizontal and edited it to be more readable.(line 200)

  1. Comment on the ‘table titles’

Take an advantage of guiding the reader by providing more descriptive details in Table titles.

Response to the comment

 Reflecting the reviewer’s suggestion, we edited the table titles in more detail as follows.

 “Table 2 : Drinking norm scores for each item of all participants”

“Table 3 : Total outcome of Drinking norms according to sociodemographic characteristics and health behaviors.”

 “Table4: Total scores for each item of attitudes toward alcohol control policies”

  1. Comment on the ‘international comparisons’

The paper could be enhanced by international comparisons, particularly in the discussion. There is significant movement in policy making. And, great success stories available. What lessons learned could be beneficial to Seoul?

Response to the comment

According to the reviewer’s comment, we added international research and implications related to drinking norms in discussion section as follows.

“Many prior studies have shown that drinking norms have been linked to drinking culture and have a significant impact on alcohol consumption[25-27]. Based on these findings, studies have shown that an intervention program developed to change one’s understanding of drinking norms can successfully reduce alcohol [28,29].”

  1. Comment on the ‘citations’

There are relatively few citations in the intro and discussion. In discussion, even sentence on alcohol harm has no citations. This needs to be addressed throughout.

Response to the comment

 According to the reviewer’s comment, we reviewed related references and added more citations.

In the discussion session, references related to alcohol harm, solitary drinking, and alcohol-control policies were cited as follows.

Lim, J.; Kim, H. Factors related to problem drinking and solitary drinking: Online survey with one person household women in early adulthood. J Korean Acad Psychiatr Ment Health Nurs 2021, 30, 30-41.

Griswold, Max G.; Fullman, N.; Hawley, C.; Arian, N.; Zimsen, S.R.; Tymeson, H.D.; Fariolo, A. Alcohol use and burden for 195 countries and territories, 1990–2016: a systematic analysis for the Global Burden of Disease Study 2016. The Lancet 2018,392. 1015-1035.

Fell, J. C. Approaches for reducing alcohol-impaired driving: evidence-based legislation, law enforcement strategies, sanctions, and alcohol-control policies. Forensic Sci Rev 2019, 31, 161-184.

  1. Comment on the conclusions

The manuscripts concludes with the limitations. Please consider adding something more thoughtful about next steps, how the data will be used or any expectations.

Response to the comment

According to the reviewers’ suggestion, we extracted the conclusion at the end of text as follows.

“In conclusion, population groups need to have drinking norms oriented towards alcohol prevention and harm reduction to increase acceptance of restrictive alcohol policies. Therefore, continuous social efforts are necessary to help groups with a positive attitude toward drinking in order for them to adopt desirable drinking norms.” (revised line)

Round 2

Reviewer 1 Report

Comments and Suggestions for Authors

Dear Authors,

Thank you for your accepted my suggestion. Thank you for corrected of the text. This manuscript include the importand the information conected with promotion of the health life style.

Best regards

Reviewer